



Exploring the role of hydrological pathways in modulating North Atlantic Oscillation (NAO)
teleconnection periodicities from UK rainfall to streamflow.
William Rust [a]; Mark Cuthbert [b]; John Bloomfield [c]; Ron Corstanje [d]; Nicholas Howden [e]; Ian Holman [a]
a Cranfield Water Science Institute (CWSI), Cranfield University, Bedford MK43 0AL
b School of Earth and Ocean Sciences, Cardiff University, Park Place, Cardiff, CF10 3AT
c British Geological Survey, Wallingford, OX10 8ED
d Centre for Environment and Agricultural Informatics, Cranfield University, Bedford MK43 0AL
e Queen's Building, University Walk, Clifton BS8 1TR
Correspondence to William Rust (w.d.rust@cranfield.ac.uk)
Abstract
An understanding of multi-annual behaviour in streamflow allows for better estimation of the
risks associated with hydrological extremes. This is can enable improved preparedness for
streamflow-dependant services such as freshwater ecology, drinking water supply and
agriculture. Recently, efforts have focused on detecting relationships between long-term
hydrological behaviour and oscillatory climate systems (such as the NAO). For instance, the
approximate 7-year periodicity of the NAO has been detected in groundwater level records in
the North Atlantic region, providing a degree of forecasting for future water resource extremes
due to their repeating, periodic nature. However, the extent to which these 7-year NAO-like
signals are propagated to streamflow, and the catchment processes that modulate this
propagation, are currently unknown. Here, we show statistically significant evidence that these
7-year periodicities are present in streamflow (and associated catchment rainfall), by applying
multi-resolution analysis to a large dataset of streamflow and associated catchment rainfall
across the UK. Our results provide new evidence for spatial patterns of NAO periodicities in
UK rainfall with areas of greatest NAO signal found in south west England, South Wales,
Northern Ireland and central Scotland, and that NAO-like periodicities account for a greater
proportion of streamflow variability in these areas. Furthermore, we show that subsurface
pathway contribution, as characterised by the Baseflow Index (BFI), and the response times
of subsurface pathways, as characterised by Groundwater response Time (GRT), are
influential factors for streamflow sensitivity to these NAO-like cycles. Our results provide





critical process understanding for the screening and use of streamflow teleconnections for the
improving the practice and policy of long-term streamflow resource management.

1.  Introduction

The North Atlantic Oscillation (NAO) is a dipolar system of atmospheric pressure in the North
Atlantic region that is known to modulate European meteorological and hydrological conditions
(Hurrell and Deser, 2010; Lawler et al., 2011; Faust et al., 2016; West et al., 2019). It has been
shown that the winter state of the NAO drives wetter or drier conditions in rainfall and river
flow in the same winter season (Uvo, 2003; Bouwer et al., 2006; Fritier et al., 2012; Riaz et
al., 2017), by modulating the westerly storm track (Trigo et al., 2002; Dawson et al., 2004) and
Gulf Stream strength (Frankignoul et al., 2001; Chaudhuri et al., 2011; Watelet et al., 2017).
As such, this teleconnection has been shown to account for the majority of European winter
water balance variability, and is particularly influential in western Europe (Alexander et al.,
2005; López-Moreno et al., 2011).

In addition to sub-annual variability, the NAO exhibits a principal multi-annual cycle of between
6 and 9 years (Hurrell, 1995; Hurrell et al., 2003; Zhang et al., 2011). Much research has
focused on detecting the propagation of these multi-annual signals to hydrological records,
given their potential to improve long-term projections of hydrological extremes (Tabari et al.,
2014; Su et al., 2017; Rust et al., 2019). To date, NAO-like multi-annual cycles have been
detected principally in groundwater level records in the USA (e.g Kuss and Gurdak, 2014),
continental Europe (e.g. Neves et al., 2019) and the UK (e.g. Holman et al., 2011; Rust et al.,
2019), in part due to the relative sensitivity of groundwater stores to long-term changes in
recharge (Bloomfield and Marchant, 2013; Forootan et al., 2018; Van Loon, 2015).
Furthermore, Rust et al (2019) compared NAO-like periodicities in composite rainfall records
and groundwater levels in the UK's principal aquifers, demonstrating the degree to which



periodic NAO teleconnection signals can be modulated through part of the hydrological cycle.
Given the presence of these multi-annual cycles in both UK rainfall and groundwater records,
it follows that these signals may be propagated to streamflow, particularly in groundwater-
dominated streams such as those found in many parts of southern and eastern England
(Bloomfield et al., 2009). High baseflow streams are often critical for the function of public
water supply, freshwater ecosystems, and provide a greater amenity value for surrounding
areas (Acreman and Dunbar, 2004). Therefore, an understanding of the catchment processes
that modulate teleconnection-driven multi-annual extremes in streamflow may provide a new
opportunity to better manage the long-term use and sustainability of these streamflow-
dependant services (Acreman and Dunbar, 2004; Chun et al., 2009). While existing studies
have shown that the winter-averaged NAO can modulate streamflow in the UK at an annual
scale (Kingston et al., 2006), the strength and spatiality of NAO-like multi-annual cycles in
streamflow, and the catchment processes that modulate them, have yet to be assessed.

Hydrological pathways are often used to conceptualise the propagation of effective rainfall
signals (rainfall minus evapotranspiration) through a catchment to streamflow (Misumi et al.,
2001; Bracken et al., 2013; Crossman et al., 2014; Lane et al., 2019). For example, surface
pathways are the result of infiltration- or saturation-excess runoff from the land surface and
provide a direct response to rainfall in the order of hours or days (Nathan and McMahon, 1990;
Gericke and Smithers, 2014; Kronholm and Capel, 2016). Subsurface pathways (such as the
travel of water through the unsaturated zone and groundwater flow paths to channel baseflow)
exhibit generally lower celerities than surface pathways and can produce a protracted
response to rainfall in the order of months or years where faster subsurface pathways
dominate (Carr and Simpson, 2018; Hellwig and Stahl, 2018), but ranging to decades or even
millennia for longer, deeper groundwater flow pathways with low hydraulic diffusivity
(Rousseau-Gueutin *et al.*, 2013; Cuthbert *et al.*, 2019). Existing research into periodic NAO
teleconnections with groundwater resources has highlighted the importance of subsurface


pathway responsiveness in modulating NAO-like signals in groundwater stores (Kuss and
Gurdak, 2014; Neves *et al.*, 2019; Rust *et al.*, 2019). Where a groundwater resource receives
a periodic recharge signal (such as those from a climatic teleconnection), Townley (1995)
suggests that pathways with response times shorter than the period length will propagate
these signals to baseflow more effectively, with minimal damping. Conversely, groundwater
pathways with response times longer than the period length cannot convey these signals to
the stream at a sufficient rate, meaning the amplitude of the periodic signal is damped as it
passes through the aquifer. Therefore, in the case of streamflow, we may expect that;
i.    the propagation of NAO-like multi-annual periodic signals from rainfall to
streamflow is dependent on the relative contribution of surface and subsurface
(e.g. groundwater) hydrological pathways within a catchment.
ii.    response times of subsurface pathways will modulate the amplitude of multi-annual
periodic signals in streamflow where they are propagated by subsurface pathways
Finally, these effects (modulation of NAO signal propagation by hydrological pathways) may
be expected to differ between winter and summer streamflow. Catchments in the UK have
been shown to receive the strongest NAO signals in winter rainfall (Alexander *et al.*, 2005;
Hurrell and Deser, 2010; West *et al.*, 2019). However, given the degree of fine-scale variability
seen in precipitation records (Meinke *et al.*, 2005), winter streamflow may contain a relatively
low signal-to-noise ratio as surface (and some subsurface) hydrological pathways respond to
rainfall within the same winter season. Conversely, slower subsurface pathways provide a
protracted response to winter rainfall signals, and are generally accepted to filter finer-scale
variability (Bloomfield and Marchant, 2013). As such, we may expect the NAO teleconnection
to have a greater influence on summer streamflow in permeable catchments which have a
greater contribution from sub surface pathways (baseflow), and proportionally less
contribution from surface pathways. In these instances, we may expect the teleconnection
between NAO and UK streamflow may be asymmetric between summer and winter. If multi-
annual periodic signals in streamflow are present via a teleconnection with the NAO, their use





for improving long-term projection of hydrological extremes will rely on an understanding of
the catchment processes that modulate the strength of these signals, and their seasonal
sensitivities.

The aim of this paper is to assess the extent to which NAO-like multi-annual signals are
propagated from rainfall to streamflow across the UK, and to assess how this is modulated by
the relative contribution of faster and slower hydrological pathways.

This aim will be met by addressing the following research objectives:

1. Characterise the strength, statistical significance and spatial distribution of NAO-like

multi-annual periodicities in rainfall and associated UK streamflow

2. Quantify the relationship between catchment pathway contribution and response times

and the NAO teleconnection by comparing NAO-like periodicity strength in summer

and winter streamflow.




2. Data and Methods

2.1.       Streamflow data

Monthly streamflow data and catchment metadata from the UK National River Flow Archive
(NRFA; Dixon et al., 2013: http://nrfa.ceh.ac.uk/) has been used in this study. Gauging stations
with more than 20 years of continuous streamflow data (and coincident catchment rainfall,
discussed in the following section), and no data gaps greater than 12 months were initially
selected. Where there were multiple gauging stations in a single named river catchment, only
the sites with the largest catchment area were taken forward. This produced a final list of 705





streamflow gauging stations for use in this study. These streamflow records range from 20 to
128 years in length, with a median length of 44.6 years (536 months). These sites provide a
representative sample of sites from across the UK, with minimal bias towards the south of
England, as indicated by Fig 1.
2.1.        Catchment Rainfall data
Calculated monthly rainfall totals for each streamflow gauge catchment are also provided by
the NRFA. This dataset has been derived from CEH-GEAR data (Tanguy et al., 2016), which
covers the 1890 – 2015 time period, using NRFA catchment boundaries. This catchment
rainfall dataset has been used in multiple studies investigating catchment hydrology dynamics
and catchment response to rainfall signals (Chiverton et al., 2015; Guillod et al., 2018; Gnann
et al., 2019).

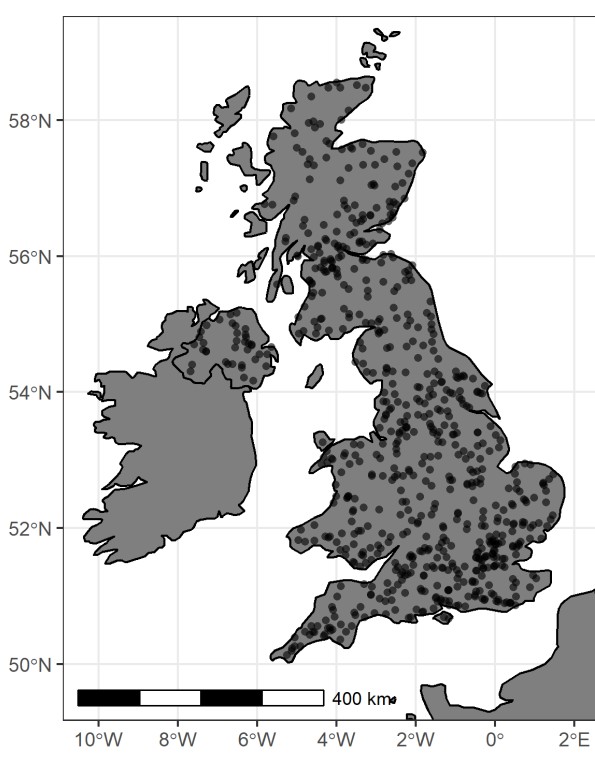


Figure 1 – Locations of streamflow gauges used in this study.







2.2.        Catchment Metadata
In order to categorise the relative influence of surface and subsurface hydrological pathways
on streamflow, the Base Flow Index (BFI) from the NRFA has been used for each streamflow
gauge (Gustard et al., 1992). The BFI is a calculated proportion of the flow hydrograph
(ranging from 0 to 1) that is derived from slower subsurface pathways such as groundwater-
driven baseflow, where 1 is entirely baseflow. While empirical, BFI has been shown to be
effective in relating physical catchment pathway processes to streamflow behaviour
(Bloomfield et al., 2009; Chiverton et al.,2015). Figure 2a shows the spatial distribution of BFI
across the UK. Higher BFI values are generally found in catchments with greater groundwater
influence, such as those in southern and eastern England which are dominated by the UK's
Chalk aquifer (Marsh and Hannaford, 2008).  Areas of moderate BFI can also be found where
there are substantial superficial or glacial deposits such as western England, central Wales
and eastern Scotland. In this study, BFI has been grouped into "Low" (0 - 0.25), "Medium"
(0.25 - 0.5), "High" (0.5 - 0.75) and "Very High" (0.75 - 1).
In addition to the BFI, the global dataset of Groundwater Response Times (GRT), developed
by Cuthbert et al (2019), has been used in this study to estimate the responsiveness of
unconfined subsurface pathways. GRT [T] can be conceptualised as a measure of the time
required for a groundwater store to return to an equilibrium after a perturbation in recharge,
and is given by:

$$\text{GRT} = \frac{L^2 S}{\beta T} \qquad\qquad \text{(Eq.1)}$$

where $\beta$ is a dimensionless constant, $T$ is transmissivity [$L^2T^{-1}$], $S$ is storativity [–] and $L$ is the
characteristic groundwater flow path length approximated for unconfined groundwater
systems by the distance between perennial streams [L]. In this study, the mean GRT was
taken for each of the NRFA catchments boundaries for each streamflow gauge. $\text{Log}_{10}$ of GRT



is displayed in Fig. 2b for clarity purposes, as for gauge catchments used in this study the
GRT ranges from approximately 1 year to approximately a million years (e.g. in very low
permeability geological formations). While the mapping of GRT was carried out using global
datasets with their inherent uncertainties, it should nevertheless enable categorisation of the
likely timescales of groundwater response sufficiently well for the purposes of this paper. GRT
is seen to be lowest (indicating shorter response times) in areas similar to areas of higher BFI;
southern and eastern England. Lower GRT values are also seen in Northern England.
Greatest GRT values are found in the south-east of England, and along the west coast of
England and Wales. While BFI and GRT appear inversely similar in spatial extent, their
correlation is low (r = -0.304). This is to be expected as they measure different aspects of
catchment process. Unlike BFI, which is an empirical measure of the degree to which slower
pathways contribute to streamflow variability (which may encompass groundwater and
throughflow), GRT is an estimate of the responsiveness of groundwater stores. In this study,
GRT is grouped into five categories: 0-4 years 4-8 years; 8-16 years; 16-32 years and greater
than 32 years.
Finally, Standard Average Annual Rainfall (SAAR) for the period 1961-1990 is also provided
as metadata in the NFRA. While not used in our analysis, it is provided here to aid later
discussion. There is a clear zonal divide in SAAR distribution in the UK with greater values on
the west coast and lower values found on the east coast of the UK and central England.
Greatest values are found in west Scotland.

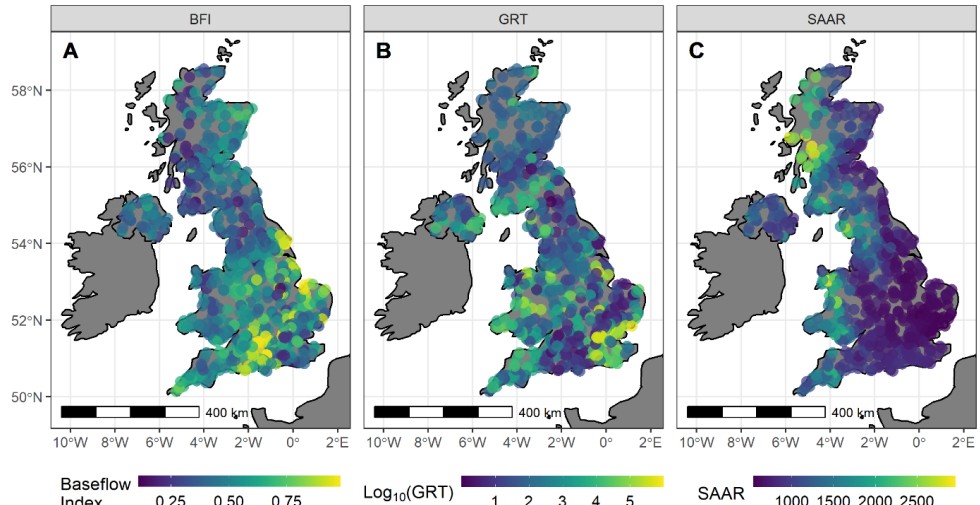

Figure 2 – Spatial distribution of a. Base Flow Index (BFI), b. $Log_{10}$(GRT) and c. Standard Average Annual Rainfall (SAAR) for each streamflow record.

## 2.3.   Methods

### 2.3.1.   Data Pre-processing

In this study we follow a similar pre-processing methodology to that set out in Rust et al (2019). The following pre-processing steps were undertaken. Firstly, all time-series were centred on the long-term mean and normalized to the standard deviation to produce a time series of anomalies. This is to allow spectra between rainfall and streamflow (and between sites across the UK) to be directly compared. From these anomalies; three time-series were created for both streamflow and rainfall, namely; monthly, winter-average (DJF) data and summer-average (JJA) data.

### 2.3.2.   Continuous Wavelet Transform (CWT) and identification of Mutli-annual periodic signals

The CWT is a multi-resolution analysis use to quantify the amplitude of periodic components of a timeseries. It has been used increasingly on hydrological datasets to extract information on non-stationary periodic behaviours in rainfall (Rashid et al., 2015), river flow (Su et al.,



2017), and groundwater (Holman et al., 2011; Kuss and Gurdak, 2014). We use the package
"WaveletComp" produced by Rosch and Schmidbauer (2018) for all transformations in this
paper. The wavelet power, W, represents a dimensionless, absolute measure of periodic
amplitude at a time index, t, and scale index, s, through a convolution of the data sequence
($x_t$) with scaled and time-shifted versions of a wavelet:

$$W(\tau,s) = \frac{1}{s}\left|\sum_t x_t \frac{1}{\sqrt{s}}\psi * \left(\frac{t-\tau}{s}\right)\right|^2$$

(Eq. 2)

where the asterisk represents the complex conjugate, t is the localized time index, s is the
wavelet scale, and dt is increment of time shifting of the wavelet. The choice of the set of
scales, s, determines the wavelet coverage of the series in its frequency domain. The Morlet
wavelet was favoured over other candidates due to its good definition in both the time and
frequency domains (Tremblay et al., 2011; Holman et al., 2011). Since all datasets have been
converted to anomalies prior to the CWT, the calculated wavelet power represents the relative
strength of periodicities within the frequency spectra of the anomaly dataset. CWT was
undertaken on all three dataset time resolutions (monthly, winter-average and summer-
average) to gain an understanding of the periodicities within UK seasonal hydrological data.

2.3.3.  Wavelet Significance Testing
Environmental datasets generally exhibit non-zero lag-1 autocorrelations (AR1) due to system
storages (Meinke et al., 2005). As a result, they can produce low frequencies as a function of
internal variance, rather than an external forcing (Allen and Smith, 1996; Meinke et al., 2005;
Velasco et al., 2015). In order to assess whether the periodicities detected as part of the CWT
are likely to be the result of noise within the dataset, a red-noise (AR1) significance test has
been carried out on all wavelet transforms. For this, 1000 randomly constructed synthetic
series with the same AR1 as the original time series were created using Monte Carlo methods.
Wavelet spectra maxima from these represent periodicity strength that can arise from a purely



red noise process. Wavelet powers from the original dataset that are greater than these "red"
periodicities are therefore considered to be driven by a process other than red noise, thus
rejecting the null hypothesis. Teleconnection processes are often noisy meaning identification
of significant periodic behaviours in hydrological datasets can be problematic (Rust *et al.*,
2019). While we highlight any periodicities equal to or above a 95 % confidence interval (CI)
(<= 0.05 p-values, due to convention), we also report the full range of p-value results in order
to accrue an understanding of periodic forcing across the large dataset.

2.3.4.   Identification of likely NAO-driven periodicities in rainfall and streamflow

An exploratory approach was undertaken to identify the most prominent multi-annual
periodicity across the streamflow records. Periods with a defined peak and greater than one
year in length were identified within the monthly streamflow spectra. Records where no defined
multi-annual peak in the wavelet power spectrum could be identified were ignored. The
maximum wavelet power within the $25^{th}$ and $75^{th}$ percentile of the identified peak-periodicities
was calculated for each of the streamflow and rainfall datasets, for all of the time resolutions.
This produced a wavelet power for each dataset that is considered NAO-like, while minimising
the influence of neighbouring periodicities. Since there is an expectation of spatially-varying
NAO-like signal strength in rainfall (Rust et al., 2018), it is necessary to minimise any
confounding correlation between streamflow and rainfall NAO signals before testing
streamflow NAO signals against catchment responsiveness. As such a residual NAO-like
wavelet power was calculated for each of the streamflow spectra by subtracting the NAO-like
wavelet power for the catchment rainfall from the streamflow wavelet power of the same site.
This therefore also acts as a measure of the modulations of signal strengths between rainfall
and streamflow. For the Summer streamflow NAO powers, a pragmatic decision was made to
construct the residual using summer streamflow and winter rainfall, given the expected low
signal presence in summer rainfall and the protracted influence of winter rainfall on summer
baseflow. (Hannaford and Harvey, 2010). It is important to note that modulation, in this case,
refers to a change in the spectral strength of NAO-like periods between rainfall and
streamflow, and not a measure of change in the amplitude of a temporally periodic behaviour
between rainfall and streamflow.
2.3.5. Testing the relationship between NAO-like signal strength and hydrological
pathway characteristics
In order to test the significance of the relationship between the BFI and GRT groups and NAO-
like signal presence, the Mann Whitney U test (MWU) was undertaken.  The MWU tests
the null hypothesis that it is equally likely that a randomly selected value from one population
will be different to a randomly selected value from a second population. We use this test here
to investigate whether populations from each successive pair of ordinal groups (e.g. Low-Med
for BFI) have significantly different distributions.
3.  Results
3.1.    Average wavelet power and p-values
Wavelet power spectra and p-values for each of the 705 streamflow and catchment rainfall
records are displayed in Fig. 3 and 4 respectively. Average wavelet power and p-values across
all sites are shown by the thick line in each plot. Wavelet power is a measure of the relative
strength of periodic behaviour (periodicity) within a dataset. In the monthly streamflow and
rainfall spectra figures, two discrete bands of periodicity can be seen in the average wavelet
powers. These are centred on the 1-year and approximately 7-year periodicity; with average
1-year wavelet powers of 0.661 (range: 0.113-0.980) for streamflow and 0.284 (range: 0.051-
0.621) for catchment rainfall; and average 7-year wavelet powers of 0.056 (range: 0.002-
0.360) for streamflow and 0.036 (range: 0.003 and 0.070) for rainfall. The ~7 year periodicity
(P7) signal is also exhibited as discrete periodicities in the seasonal data; with mean P7
wavelet powers of 0.274 (0.029 – 0.582) and 0.198 (0.010 – 0.571) for winter and summer
streamflow; and 0.253 (0.015 – 0.472) and 0.107 (0.006 – 0.535) for winter and summer
catchment rainfall respectively.
These strengths are generally reflected in the wavelet p-values, with bands of lower p-values
at the 1 and ~7 year in monthly data, and ~7 year in the seasonal data. Wavelet p-values
indicate the likelihood that the detected wavelet powers are not the result of external forcing.
As such, lower values indicate increased significance of external forcing over the red noise
null hypothesis. Wavelet p-values are generally lower in the monthly catchment rainfall spectra
(0.002 – 0.996; mean of 0.289), compared with monthly streamflow (0 – 0.995; mean of 0.443),
but this may be an artefact of longer autocorrelations in groundwater records relative to rainfall.
Wavelet p-values are comparable for the seasonal spectra, with the exception of summer
rainfall which shows the lowest significance; (winter rainfall; 0.003 – 0.995 (mean of 0.148);
winter streamflow; 0.001 – 0.839 (mean of 0.129); summer rainfall; 0.005 – 0.992 (mean of
0.462); summer streamflow; 0.000 – 0.997 (mean of 0.348)). Summer rainfall shows the
weakest wavelet powers and greatest p-values for the P7 band.

Discrete bands of decreased average wavelet p-values can also be seen between 16-32 years
for all the streamflow (monthly: 0.502, winter: 0.400, summer: 0.209) and rainfall datasets
(monthly: 0.456, winter: 0.569, summer: 0.355). This periodicity band however exhibits
negligible average wavelet power indicating minimal influence on variability. In the winter- and
summer-average power spectra there is a band of increased strength at the 2-3 year
periodicity.  In the winter-average data there is no comparably low p-value, suggesting these
higher powers are the result of noise within the averaged time series However, all the summer
spectra, appear to exhibit some decreased p-value at this 2-3-year band.

3.2.      Spatial distribution of Wavelet Powers
The main multi-annual periodicity detected in the winter and summer river flow data (~7 years)
was mapped for seasonal catchment rainfall and streamflow in Fig. 5. The winter spatial
distributions show three distinct areas of increased wavelet power and significance, shared



between catchment rainfall and streamflow. The largest area is located in the south-west of
England and south Wales, extending north into the Midlands and east into the south east of
England in the streamflow data. For rainfall, this area encompasses 101 of the 221 catchments
with significant (greater than 95% CI) P7 wavelet power, and 224 of the 262 significant sites
in streamflow. The two other areas of increased wavelet significance in rainfall and streamflow
cover Northern Ireland (20 significant sites for rainfall; 12 for streamflow) and central Scotland
(30 significant sites for rainfall; 25 for streamflow). There are also stronger P7 wavelet powers
along the west coast of the UK in both winter rainfall and streamflow, however most significant
powers (> 95% CI) are found in England and Wales. Additionally, the location of the greatest
wavelet powers differs between winter rainfall and streamflow. Winter rainfall shows higher
wavelet powers along the south-west peninsula of England, and south Wales, whereas the
greatest winter streamflow wavelet powers are found in South and south-eastern England and
appear to be co-located over the Chalk and other principal aquifers (Allen et al., 1997).
Little spatial structure exists in P7 wavelet power and significance for the summer-average
rainfall data. Some increased density in significance is seen towards the south coast of
England; however, this may be due to the increased density of sites in this region as seen in
Fig. 1, especially given the negligible average P7 wavelet strength displayed in Fig. 3.
Conversely, summer-average river flows show some clear spatial structure of wavelet power
and significance, in the South of England, where 51 of the 70 sites with significant P7 powers
are located. Again, these sites appear to be co-located over the Chalk aquifer (Allen et al.,

334   1997).


3.3.      Testing of hydrological pathways
Figure 6 shows scatter plots of the P7 residual wavelet powers (RWP) for winter and summer
streamflow plotted by BFI category (Fig. 6a), and a comparison of median P7 RWP with
significance results from the MWU tests (Fig. 6b).  Winter P7 median RWPs show a trend of





increasing wavelet powers with increasing BFI category, with the exception of between the
Low and Medium categories (0.001, -0.002, 0.019 and 0.093 for Low, Medium, High and Very
High groups respectively). A similar relationship is seen in the Summer median P7 RWPs (-
0.063, -0.079, -0.054 for Low, Medium and High groups), with a notably steeper increase for
the final group when compared to winter P7 residuals (increasing to 0.101). This brings the
median P7 residual powers for summer streamflow to a comparable magnitude to winter
streamflow. In general, winter median P7 residual powers are close to zero except for the Very
High category, indicating minimal modulation of P7 signal strength between rainfall and
streamflow in the catchments with Low to High BFI. Summer P7 residuals are negative for
Low – High BFI catchments indicating a reduction in P7 wavelet powers in streamflow
compared to winter rainfall. The median P7 residual for sites in the Very High BFI is the only
positive residual for summer streamflow, indicating an increase in relative P7 signal strength
between winter rainfall and summer streamflow for these sites.
Figure 7 shows P7 RWP plotted against Groundwater Response Times (GRT) groups showing
all gauges (Fig. 7a), and median RWP with significant results from the MWU tests (Fig. 7b).
Winter streamflow shows higher, positive median RWP across all GRT groups (0.056, 0.079,
0.017, 0.009, 0.002, for the 0-4, 4-8, 8-16, 16-32 and 32+ year groups respectively), whereas
summer streamflow only shows positive RWPs for catchments in the 0-4 and 4-8 year GRT
groups (median RWP of 0.014 and 0.024 respectively). GRTs groups greater than or equal to
8 years all show negative median RWPs (-0.011, -0.058 and -0.074 for 8-16, 18-32 and 32+
year groups respectively). Both winter and summer streamflow show decreasing median
RWPs with increasing GRT, with the exception of the 4-8 year GRT group, which shows the
greatest median RWP in both winter and summer. Significant difference between GRT groups
are found between 0-4 and 4-8, and 4-8 and 8-16 for winter streamflow, and between 4-8 and
8-16 for summer streamflow.



Figure 3 - Stacked streamflow wavelet spectra power (left) and p-values (right) from normalised Monthly, Winter and Summer resolution data of 705 catchments. 95% Confidence interval is shown as a dashed black line on the right column figures. Opacity of each average spectra line has been lowered to allow general trends to be identified.


Figure 4 – As Fig. 3 but for catchment rainfall data.


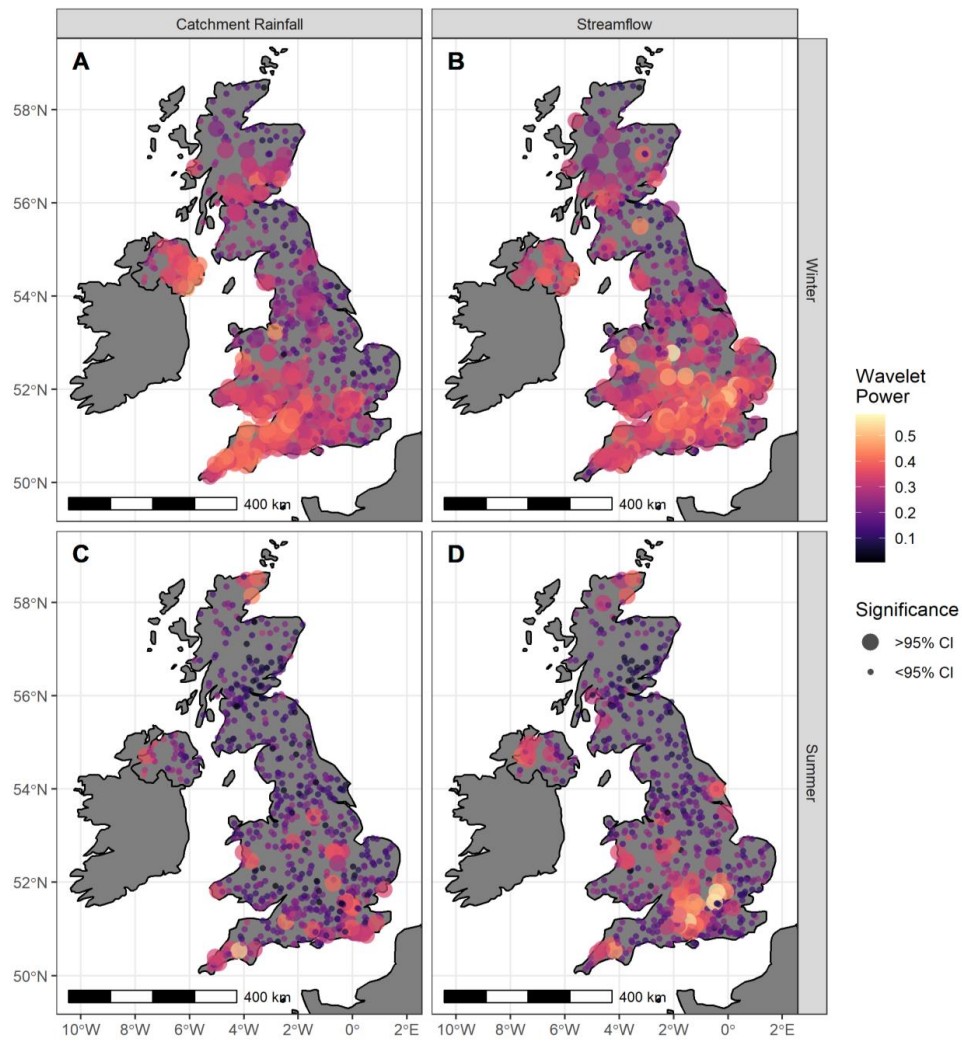


Figure 5 – Spatial distribution of ~7-year periodicity wavelet power and significance in catchment rainfall and streamflow, for winter and summer-averaged datasets.





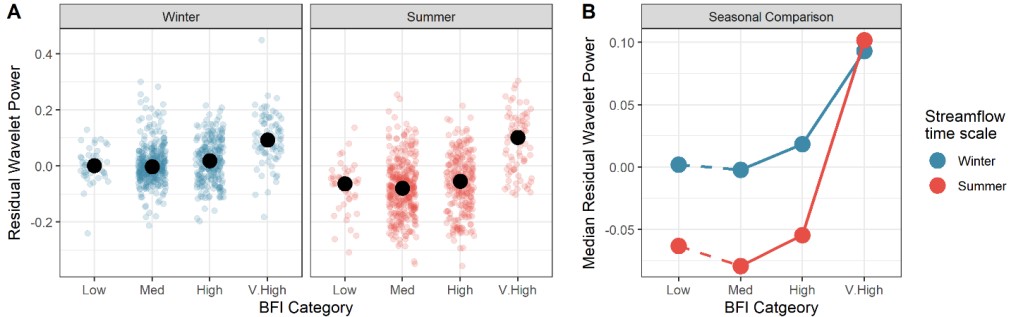


Figure 6 – A) shows jittered scatter plots for residual wavelet powers in winter and summer, categorised by BFI; bold black points mark the average residual wavelet power for each BFI category. B) compares these median residual wavelet powers with significant changes between groups shown as solid lines, and non-significant changes between groups shown in dashed lines.


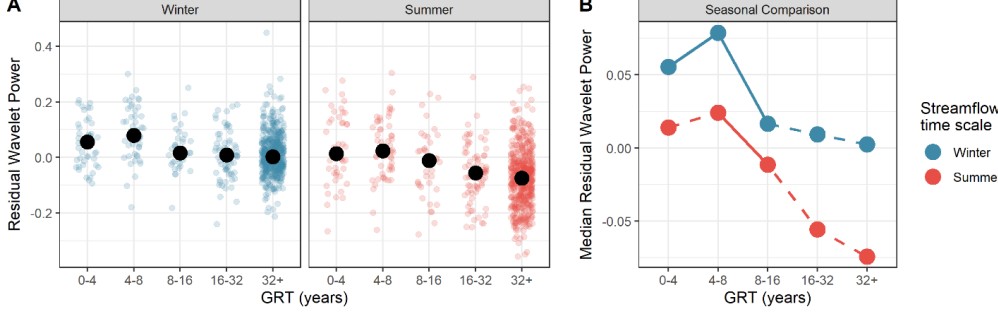


Figure 7 – as figure 6 but for the Groundwater Response Times (GRT)
















## 4. Discussion

### 4.1. Detecting a teleconnection between the NAO and UK Streamflow

Our results indicate that the dominant multi-annual periodicity in UK streamflow (and catchment rainfall) is that of an approximately 7-year cycle. This cycle closely compares to the principle 7-year periodicity documented in the strength of the NAO's atmospheric dipole, which has been associated with multi-annual periodicities in hydrometeorological records globally (Rust et al 2019; Meinke et al., 2005; Tremblay et al., 2011; Kuss and Gurdak, 2014; Holman et al., 2011; Neves et al., 2019). We show here that this ~7 year cycle is wide-spread within rainfall and streamflow variability across the UK, with the majority of streamflow and rainfall records assessed here exhibiting a coherent band of increased periodicity strength and significance around this 7-year frequency range. This, combined with greater significance for this periodicity, indicates an external control on this multi-annual mode of variability. As such, we build upon evidence in existing research that documents the teleconnection between the NAO and rainfall in Europe and show new evidence of the propagation of the NAO's ~7 year cycle to UK streamflow variability. Additionally, we detect expected differences between signal presence in summer and winter rainfall, showing the majority of NAO-like signals are present during the winter months, and absent in the summer. This generally agrees with existing research showing that NAO's control over European rainfall is primarily expressed in winter months (Trigo et al., 2004; West et al., 2019).

Olsen et al (2012) show that the NAO also exhibits a shorter (and weaker) 2-3 year periodicity between winter and summer NAO index values, and a ~6 year periodicity in summer index data. There are several other periodicities apparent in both our streamflow (Fig. 3) and rainfall (Fig. 4) wavelet power and significance, with peaks at ~2 years and ~5 years in the summer data. Therefore, it is possible that our results show that UK streamflow is driven by multiple periodicities in the NAO. Finally, our results (Figs. 3 and 4) indicate a 16-32 year periodicity in all the wavelet p-values which has previously been associated with the East Atlantic Pattern (EA) (Holman et al., 2011; Rust et al., 2019). As a secondary control on winter rainfall (after





the NAO) (Wallace and Gutzler, 1981), this would explain the weaker strength of the 16-32
year cycle compared the NAO-like ~7 year cycle shown in the results. For the remainder of
the paper, we will focus on the propagation on the NAO's principal periodicity (~7 years) that
we have detected in both monthly and seasonal datasets.

4.2.      Controls on NAO-like signals in catchment rainfall
We provide new evidence that the influence of the NAO's ~7 year periodicity on UK rainfall is
seen in the winter months, and is heavily localised to the Southwest of England, South Wales,
the east coast of Northern Ireland and central band Scotland (Fig. 5). This is contrary to
previous research that has typically found strongest relationships between the NAO Index and
UK rainfall along the west coast of the UK, particularly the west coast of Scotland (Murphy and
Washington, 2001; Fowler and Kilsby, 2002; West et al., 2019). Rust et al (2018) suggests
that the NAO's control on UK rainfall variability may operate through two teleconnection
pathways; atmospheric and oceanic, both with different temporal sensitivities. For instance, a
positive (negative) NAO is known to drive increased (decreased) strength in the westerly storm
tracks at an instantaneous timescale (Dawson et al., 2002; Walter and Graf, 2005), which
subsequently drives wetter conditions on the west coast of the UK during the winter months
(Walter and Graf, 2005).   As such, we may expect methodologies that assess the
instantaneous relationship between the NAO and rainfall (such as existing research) to show
sensitivity to this atmospheric teleconnection route (e.g. Westerly Storm tracks). Conversely,
the NAO is also known to increase the strength and meridional tilt of the Gulf Stream (Taylor
and Stephens, 1998; Gangopadhyay et al., 2016; Watelet et al., 2017) which, as an oceanic
process, filters out finer-scale variability and is sensitive to control at multi-annual timescales
(Hurrell and Deser, 2010). It follows, therefore, that the Gulf Stream may be more sensitive to
the NAO's long-term influence (such as its principal ~7 year periodicity) and relatively
insensitive to its finer-scale variability (Rust et al 2018).  Haarsma et al (2015) have shown
that the Gulf Stream is particularly influential on rainfall variability in the South East of England





through its modulation of sea surface temperatures. As such, the increased strength in NAO-
like signals in winter rainfall shown here in the South East of England may be an expression
of the NAO's long-term control on the Gulf Stream.

4.3.      Hydrological drivers for signal strengths
We have shown that NAO-like periodicities are localised to specific regions in the UK in winter
rainfall (Fig. 5a) and are negligible in summer rainfall (Fig. 5c). This suggests that NAO-like
periodicities in summer streamflow do not originate from summer rainfall, and that catchment
processes that drive winter rainfall signal propagation to summer streamflow (e.g. subsurface
pathways (Haslinger *et al.*, 2014; Folland *et al.*, 2015; Barker *et al.*, 2016)) may inform our
understanding of catchment controls on the NAO teleconnection with streamflow. Here, we
provide statistically significant evidence that periodic NAO-like signals in rainfall are
propagated to streamflow differently between winter and summer months, depending on the
contribution from different hydrological pathways (and their response times). Furthermore, we
provide evidence that pathways of specific response times propagate NAO-like periodic
signals to UK streamflow more effectively than others, highlighting the catchment properties
that may produce a sensitivity to the NAO teleconnection with streamflow. Below, we discuss
how these relationships align with current hydrological understanding.
Rust et al (2019) establishes that multi-annual NAO-like periodicities in groundwater level
records are considerably stronger than those in co-located rainfall records. Groundwater
behaviour generally exhibits longer autocorrelations than rainfall with negligible fine-scale
variability (noise), due to the damping effect of subsurface hydrological pathways (Townley,
1995; Dickinson, 2004; Gnann *et al.*, 2019). As such, groundwater can express a greater
signal-to-noise ratio for low frequency variations (such as those produced by the NAO
teleconnection) (Holman *et al.*, 2009; Rust *et al.*, 2018). By comparison, rainfall (which
generally contains more fine-scale (hourly – daily) variability), exhibits a lower signal-to-noise


ratio which supresses the proportional strength of multi-annual NAO-like signals (Meinke *et*
*al.*, 2005; Brown, 2018). A parallel can be drawn here with hydrological pathway influence on
streamflow, as surface pathways more closely reflect rainfall variability and subsurface
pathways more closely reflect groundwater variability (Ockenden and Chappell, 2011;
Kamruzzaman *et al.*, 2014; Mathias *et al.*, 2016; Gnann *et al.*, 2019).
Streamflow driven primarily by surface processes (e.g. BFI < 0.5) exhibits close-to-zero
median RWP in winter (Fig. 6b), indicating surface pathways affect minimal modulation of
NAO periodicity strength from winter rainfall to winter streamflow; likely due to their relatively
short response times (minutes to days) (Mathias *et al.*, 2016). This also explains why the
spatial footprint of NAO-like periodicities in winter streamflow (Fig. 5b) generally matches that
of winter rainfall (Fig. 5a) as a greater proportion of surface pathway are active in response to
greater in-season rainfall (due to more infiltration- or saturation-excess runoff from the land
surface) (Ledingham et al., 2019). Summer streamflow, where driven by surface pathways,
exhibits a damping of NAO periodicities from winter rainfall (negative median RWPs), and no
clear spatial structure of NAO-like periodicities. This is to be expected, given relative weakness
of the NAO teleconnection with UK summer rainfall (as noted by Alexander *et al.*, (2005);
Hurrell and Deser, (2010); West *et al.*, (2019), and indicated here) and the relative paucity of
subsurface pathway contributions which can protract winter rainfall signals into summer
months (Barker *et al.*, 2016). As a result, there are limited mechanisms to convey winter NAO
periodicities to summer streamflow. Conversely, streamflow that is dominated by subsurface
pathway influence (e.g. BFI > 0.75) exhibits the greatest NAO periodicities (Fig. 6b). We also
see significant increases in NAO periodicity strength with increasing BFI in all but between the
lowest two BFI categories (Low – Med). We therefore confirm our expectation that NAO
periodicities in groundwater are propagated to streamflow via subsurface pathways. This
relationship is also seen in the spatial footprints of NAO periodicities in winter (Fig. 5b) and
summer streamflow (Fig. 5d). Gauges with the strongest NAO-like periods in summer and
winter streamflow are found in catchments that are within, or that drain, the Chalk outcrop in



south central England. These catchments are known to be heavily driven by groundwater
behaviour (Marsh and Hannaford, 2008). In Fig. 5b we see the spatial footprint of NAO
periodicities in summer streamflow is localised to these Chalk-dominated catchments.
Permeable catchments such as those on the Chalk aquifer are known to slowly respond to
winter rainfall at a seasonal timescale (Hellwig and Stahl, 2018). As such, these catchments
have sufficient subsurface pathway contribution to protract NAO periodicities in winter rainfall
through to summer streamflow. Conversely, Fig. 5 also show some areas of the Chalk with
relatively low NAO-like periods, such as the southern coast of England. Similarities can be
seen here with Marchant and Bloomfield (2018) who identify discrete regions of groundwater
level behaviour within the chalk aquifer, with varying autocorrelations. The Chalk of the south
coast of England tend to have thinner superficial deposits and negligible glacial deposits
(unlike those in the area of the Chalk outcrop), producing a faster recharge response to rainfall
with shorter autocorrelations (Marsh and Hannaford, 2008; Marchant and Bloomfield, 2018).
Dickinson *et al.*, (2014) highlights the importance of unsaturated zone thickness in modulating
periodic signal progression, which may explain why catchments in the southern Chalk exhibit
lower signal-to-noise ratios for NAO periodicities.
While the relationship between NAO periodicities and streamflow BFI indicates the importance
of subsurface pathway contribution to teleconnection strength, properties of the subsurface
pathways themselves are expected to modulate periodic signal propagation from rainfall to
streamflow (Rust *et al.*, 2018). We show streamflow in catchments with shorter Groundwater
Response Times (GRT) exhibit stronger NAO-like periodicities, but the strongest NAO
periodicity is found in catchments with GRTs between 4 and 8 years. Townley (1995) shows
that where the groundwater response time of a subsurface store is longer than a periodicity in
recharge, the system will exhibit larger periodic variations in groundwater head but greater
attenuation of periodic discharges at a streamflow boundary. This is because the pathway
cannot equilibrate the periodic recharge to its hydraulic boundaries at a sufficient rate.
Conversely, where the pathway response time is shorter than that of a periodicity in recharge,





groundwater discharge will show greater periodic variations as the entire pathway is able to
convey this signal. This may explain the reduction in NAO periodicities seen as GRT increases
in Fig. 6b. Where subsurface pathway response times are longer than the principal ~7 year
periodicity of the NAO, we may expect the pathway to dampen the signal propagation to
baseflow (Townley, 1995; Dickinson, 2004). However, this process fails to explain the reduced
NAO periodicity strength seen in our results where GRT is less than the ~7 year NAO
periodicity (seen principally in the winter streamflow data). As suggested by Najafi *et al.*,
(2017) and Wilby, (2006), faster pathways can exhibit a weaker signal-to-noise ratio, when
compared to slower pathways which are known to smooth signal propagation (Barker *et al.*,
2016). As such, streamflow in catchments with the shortest GRT (i.e. 0-4 years) may exhibit
greater response to finer scale variability in rainfall which supresses the relative strength of
the NAO periodicity. This would also explain why summer streamflow does not show a
similarly reduced NAO-like period strength for the 0-4 years GRT band, as summer streamflow
generally would be expected to exhibit greater signal-to-noise ratios due to a greater
proportion of slow pathway contribution. As such, our results suggest that, in addition to the
described periodic signal modulations in Townley et al (1995), there is an ideal range of
subsurface pathway response times that are long enough to produce a greater signal-to-noise
ratio, but sufficiently short that there is minimal damping.

These results may have important implications for streamflow management, as we show that
readily available estimates of BFI and GRT may be used to screen or identify catchments
where teleconnection-driven multi-annual variability may be used to better inform risk
estimation for hydrological extremes. This is particularly important for summer streamflow
where streamflow services are often vulnerable to drought conditions (Visser *et al.*, 2019).




4. Conclusions
This paper assesses the degree to which the principal multi-annual periodicity (~7 years) of
the NAO is present in streamflow and catchment rainfall records using the Continuous wavelet
transform to identify multi-annual periodicities. We provide new evidence for the role of
oceanic and atmospheric pathways in propagating NAO periodicities to catchment rainfall, by
identifying spatial patterns of statistically significant NAO-like periodicities in UK catchment
rainfall and streamflow. This may help further explicate the varying spatial extent of the NAO
influence over Europe and the North Atlantic Region. Furthermore, we identify specific
streamflow catchment characteristics that are most responsive to the NAO periodicities in
catchment rainfall. We find that streamflow that is driven predominantly by subsurface pathway
contributions often exhibit greater NAO-like periodicities, and that subsurface pathways with
response times comparable in length to the ~7 year periodicity of the NAO produce the
greatest sensitivity to the NAO teleconnection. These findings build on the fundamental
understanding of periodic signal propagation through hydrological pathways and can be
applied to streamflow catchments globally to identify areas of greater climatic teleconnection
sensitivity. The ability to screen catchments for their potential teleconnection-driven multi-
annual variability may have direct implications for water management decision making. For
example, the permitting of surface water abstractions and their implications for ecologically
sensitive streamflow systems. Such information may help to protect vulnerable habitats or aid
appropriate investment in surface water abstraction infrastructure. Our results here make
crucial steps towards a greater understanding of how climatic teleconnections can be used to
improve water resource management practices.






**Data availability.**

The streamflow and precipitation data as well as the metadata used in this study are freely available at the NRFA website (http://nrfa.ceh.ac.uk/).

**Author contributions.**

WR designed the methodology and carried them out with supervision from all co-authors. WR prepared the article with contributions from all co-authors.

**Competing interests.**

The authors declare that they have no conflict of interest.

**Acknowledgements.**

This work was supported by the Natural Environment Research Council (grant numbers NE/M009009/1 and NE/L010070/1) and the British Geological Survey (Natural Environment Research Council). JB publishes with the permission of the Executive Director, British Geological Survey (NERC). MOC gratefully acknowledges funding for an Independent Research Fellowship from the UK Natural Environment Research Council (NE/P017819/1). We thank Angi Rosch and Harald Schmidbauer for making their wavelet package "WaveletComp" freely available.

**Financial support.**



This research has been supported by the Natural Environment Research Council (grant nos.
NE/M009009/1 and NE/L010070/1), and MOC has been supported by an Independent
Research Fellowship from the UK Natural Environment Research Council (NE/P017819/1).

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
