# Peer review of "Exploring the role of hydrological pathways in modulating multi-annual climate teleconnection"

_Hydrology and Earth System Sciences, 2020_

## Referee Comment (RC1) · Anonymous Referee #1 · 14 Oct 2020

In this paper the authors draw upon an extensive UK hydro-climatological data-set, comprising >700 gaged records of streamflow and estimated precipitation to examine 7-year periodicities related to the NAO. The range in catchment responses (and potential additional cycles at 2 and 5 years) are interesting and I think the paper is appropriate for this journal (with suggested revisions noted below), although I'm not personally convinced that this paper provides 'critical process understanding' as suggested in the abstract, and its not clear how this work can translate to improvement of the practice and policy of water resources management (as suggested in the abstract).

[Figure]

For a non-UK reader it would be useful to provide more context in introducing the work and commenting on confounding influences (e.g. there is no mention / discussion of abstraction, nor the potential for anthropogenic water use / return flows, to influence streamflow). Moreover, the authors only mention one aquifer (the Chalk) and it is difficult to interrogate the results to consider the potential for varying periodicities on other geologies (and catchments with differing land use), given the scale of figures such as Fig. 5. At this scale of analysis, I would also question whether the authors are able to infer hydrological pathways with confidence (although the Groundwater Response Time concept appears very interesting).

In addition to these thoughts, the authors may wish to consider the following points when revising the paper:

* There are multiple (>10) references missing in the ref. list including: Bloomfield & Marchant, 2013; Dixon et al. 2013; Forootan et al. 2018; Haarsma et al. 2015; Tanguy et al. 2016.

* It would be useful to include more specific details in the abstract.

* A stronger argument to demonstrate the effectiveness of the BFI in relating 'physical catchment processes to streamflow' would be helpful (e.g. as I understand the Bloomfield et al 2009 study, cited in support, focused on the Thames catchment).

* Lines 179-183 should be reworked (e.g. GRT is lowest in southern and eastern England, but highest in south-east England?).

* Figures 2 & 5: are rather small, and it would be useful to reproduce these at a larger scale.

* Is it possible to combine Figures 3 & 4 so the plots can be compared more easily?

* The Discussion is rather long and would benefit from a more selective review of the results, with more attention to suggestions for future research.

---

## Referee Comment (RC2) · Anonymous Referee #2 · 11 Nov 2020

**Review of Rust et al. for HESS**

This paper aims to explore relationships between periodicity in the NAO and UK river flows. It uses wavelet analysis and then examines the propagation of observed, apparent cyclicities from meteorological inputs to river flow response, and analyses the contribution of catchment characteristics (specifically BFI and a groundwater response variable). It builds well on previous work on wavelet applications in groundwater records by the team. The paper is well organised and clearly presented.

I am no specialist in wavelets but the analysis appears appropriate and well executed – although I have raised some potentially important questions. There is also a novelty to this work – there have been a few papers on cyclicities in UK river flows (that are not cited here: Sen, 2009; Franco Villoria et al. 2012) but this has not been studied extensively, nor has there been the consideration of hydrological propagation pathways. The work has suitable novelty and potential international interest for publication in HESS.

I do struggle to fully grasp the significance of the work, however. The wavelet analysis reveals intriguing patterns but these periodicities are difficult to interpret. I find this paper like many similar wavelet/frequency papers in that a lot is down to the interpretation of the plots and assumptions made on driving processes  - there is a lot of faith here seems to be on 'NAO like' signals which are taken and run with but to me are far from clear in the plots, and wrt previous work. There is no actual statistical link with the NAO (cf. FrancoVilloria, etc) which should be made clear from the outset (possibly even title?).

As far as I understand the NAO literature it is somewhat debatable as to whether such patterns are physically meaningful. They do crop up in some papers but I have not seen so much that is convincing as to plausible physical interpretations (this can be said about a lot of work on cyclicities in hydroclimatic series in general). Most studies indicate that if NAO cyclicity exists it is weak, sensitive to choice of NAO index, season (DJF, DJFM) and tends to come and go over time as noted in the original Hurrell paper (these non-stationaririties being another barrier to practical applications; see also Franco-Villoria on nonstationarity of NAO-flow linkages). This should be elaborated on. See also important statistical work of Mills (2004).

There is some discussion of some of this literature in the current paper (a lack of references in the reference makes it hard to find some of the key papers cited in relation to NAO periodicity) but I find some claims quite tenuous as to what the cited literature reveals wrt the findings of Rust et al. The discussion is very wide ranging, but at times highly speculative and overreaches beyond the analysis made here, and as noted below sometimes reflects misinterpretation of existing papers, or reference to work that is not directly relevant.

Finally, I am not sure of the practical significance of such findings. The authors suggest this reveals a 'degree of forecasting' and 'critical process understanding' but I am not sure how water managers can really use such information beyond a general raised awareness. And for that, there are already operational physically based seasonal to decadal prediction systems (the Met Office DePreSys system bein one) which already give an indication of potential NAO states (and moreover, more importantly, the large-scale variables that drive the NAO, like ENSO, QBO etc which are much more predictable). How would findings really be used in reality? Perhaps the authors could comment. That said, there are wider applications of these findings in explaining general time series evolution (trends and variability) a huge area of interest in the literature.

The added interest in this paper is the catchment propagation but while this element of the paper is well executed (albeit with some questions below) I am not sure it tells us much new? We know that BFI modulates precipitation signals anyway, and there's a reasonable handle on lag times from past literature, so the results are not that surprising. Great to see the GRT as a predictor though.

In summary I think the paper is novel, well executed and potentially a valuable contribution of the literature, but its framing could be improved, as well as the coverage of the significance of the findings. There are some moderate technical and presentation issues which also need to be addressed before publication in HESSD.

**Specific Comments**

L18 – 'degree of forecasting' – I see this word is being used in a general sense but I don't think this is really what the paper is offering up, rather some general tendencies of multiyear river flow behaviour in time series, itself possibly useful but not really 'forecasting', although perhaps for general preparedness. I think any findings from studies like this are more useful in providing context for studies of long-term trends and variability (helping shed light on widely reported 'flood poor', 'drought rich' periods and so on) rather than providing any information on preparedness. The authors could comment on this potential application too.

L42 – L44 (around) maybe also worth clarifying early on that the NAO is primarily a driver of wintertime variation, noted later but should be in the intro.

L48 – [and into discussion]. I wanted to look into past research on claims of NAO cyclicities but found it a bit lacking in these papers – Tabari doesn't really look at muliannial cycles; Su found cycles but did not appear to look at the NAO. Neves relevant and useful. Kuss and Gardak not in reference list; Meinke looked at ENSO; no reference in list for Tremblay; no reference in list for Olsen, but found it and appears to be a different beast on Paleo timescales so not sure of its relevance here. It's difficult to examine the wider evidence based with these missed references. Some that are cited appear to be not entirely directly comparable – I'd recommend a careful re-reading and re-positioning of this work with the literature here (and in the discussion]. See also missed references in intro and other useful international papers e.g. Labat (2010) on various possible mechanisms of cyclicity globally.

L67, L68 – there are many more studies that look at NAO influences on UK streamflow in winter and more generally and these shed important light on regional patterns in some detail, and also catchment characteristics – would be worth referring to these. See e.g. Laize et al. 2012, Svensson et al. 2015 and references in both. Laize & Hannah really looked at this propagation question in some detail (for interannual rather than periodic behaviours but still relevant).

L136 – L139. At 705 catchments, this will inevitably be a very mixed set of catchments subject to all sorts of human influences. Ideally hydroclimate studies linking large-scale drivers to hydrological response should use relatively undisturbed catchments - human disturbances can alter the chain of propagation from signal to streamflow response. I agree however that there are few such catchments (see e.g. Harrigan et al. 2018) which would limit the range of BFI/GRTs, so using the wider set is reasonable. While I expect that the outcomes from the broad-scale national picture is unaffected, there will likely be significant heterogeneity in the catchment results. This warrants a comment.

L163. This categorisation into even BFI classes will lead to very uneven numbers in the groups, given how skewed (and slightly bimodal) the BFI distribution is. Being that there is nothing meaningful about these arbitrary thresholds why not try and make them cover the distribution better? There are very few catchments at BFI <0.25, and not so many >0.75, with the great majority in the middle two classes. (I do not have this to hand but there are NRFA BFI distributions available somewhere in the literature I am sure and it would be easy to check). How would a more even classification effect the later results?

L184. The point here is a valid one that BFI is simply an empirical property from the baseflow, but this could be generalised – rather than referring to throughflow, it can reflect any terrestrial storage in soils, lakes (lake and reservoir catchments also have high BFI) and so on.

L187 – I do not know how GRT is distributed but mirroring my BFI Comment above can the authors comment on this? It looks like it is very uneven from Fig 7 with many more in the highest class.

Fig2 caption – might need to explain why this is log GRT as you have not introduced that yet, only referred to the categories.

L204 – just a comment really for future work. This study does not consider the transition seasons, which is fine. But if the focus is really on propagation from winter NAO met signals I would have thought looking at spring would be really interesting – it may help better explain the propagation from winter.

L250. (and 2.3.4 generally). I'm not sure I fully follow the logic here so it needs some clarification – at the moment it sounds like this process is doing some heavy lifting. I failed to follow the process leading up to L250 that indicates "this produced a wavelet power for each dataset that is considered NAO like'. Why would this be considered NAO like a priori? This sounds particularly important given the 'NAO like' signal is then used to producing the residual series that is so important thereafter for capturing the 'measure of modulation of signal strength'.

L258. Following the above, it also appears to be a big assumption to produce this residual series for the summer based on winter rainfall. While in very high BFI catchments a very long lag time may be expected a priori, this is not necessarily the case in many (most?) cases (See my earlier comment about the spring season being ommitted).

I'm just concerned that two comments (while partly no doubt reflecting my lack of understanding of the process) are reflective of some major assumptions being made in this study which are (at face value) in danger of baking in some of the conclusions, somewhat – hopefully a clearer exposition will help allay these fears.

L271 - Given this pairwise testing between groups, is my earlier comment about the irregular distribution of BFI a potential issue (would this look different with different more representative groups?)

L285 I am no expert on wavelets. But when I look at Figs 3 and 4 I wouldn't say a c.7 year cycle leaps out at me – rather, higher powers at a range of years <c.8 years. Especially for winter rainfall. That is, I see there are peaks in significance but are you really that confident in there being a (even approx.) 7 year cycle in these? IN fact I don't really see the 'two discrete bands of periodicity' (l279). I guess this is all down to interpretation but this causes some modest concern if this is the basis of the identification of 'NAO like' signals – please elaborate on this. It's also really difficult to see the variability in the wider cloud of catchments shaded grey, but it looks like there is a range, especially for rainfall – some comment of this would be useful.

L290. 'Wavelet p values indicate the detected wavelet powers are not the result of external forcing'. Is this strictly true, I thought this just indicates it is not AR1 generated – I assume it does not rule out that it is internal variability, which could be driven by all sorts of long-term persistence processes (see the extensive work on the Hurst phenomenon and many papers of Demetrios Koutsoyiannis), as opposed to external forcing. (corollary to this, re: my point in the general intro about physical significance, when I backtrack into the literature on NAO periodicities, back to the Hurrell papers cited, it seems far from clear as to being settled whether NAO periodicities, as they are, are externally forced or internal variability).

L294 – should this say 'river flow records' instead of groundwater?

L340 onwards and Fig 6 – note whether previous question of BFI distribution has any impact on these findings?

L416 – I would not have said this strong conclusion on the difference between the periodicities between winter ('present') and summer ('absent') rainfall really emerges from Figs 2 and 3 as noted earlier. I may be missing something but this seems quite an open interpretation of those data. Important as the seasonal differences are majored on.

L420 – L425. Following on from this, this discussion hinges on there being multiple periodicities at different peaks between the seasons, but my reading of those plots makes it hard to really pick out any of these as 'peaks'.

L425 – I also think this 16 – 32 periodicity is very difficult to see, let alone link to the EA pattern – and I could find no work on this in the Rust et al. 2019 paper cited? An important general point though is that while the NAO is the leading mode of variability there is a whole laundry list of (interacting) influences (Scandinavia pattern, AO, etc) as well as the lower frequency SST drivers (AMO, ENSO) that are not considered here given focus on NAO-like, even though other work suggests they could also manifest themselves on similar timescales (see e.g. Labat, Villoria).   Worth comment in intro & discussion?

L453. Haarsma not in the reference list, But on looking at this paper I don't see this SE England outcome on any of their results maps? Please clarify. In general, I find this whole gulf stream section really speculative. I looked (admittedly quickly) but could not really find much in these papers to support this e.g. concurrent but lagged correlations in Watelet rather than periodicities; little specific mention of GS in Hurrel and Deser. A lot is made of the GS as a mechanism for the key NAO-like behaviour central to this study, so this reference to other work is important and could be checked and strengthened.

L576 – L582. Related to my points in the intro, good to see potential applications but this is quite a long way off from what is discovered in this paper so some of these claims could be moderated.

**References**

Franco-Viloria et al. 2012. http://www.jenvstat.org/v03/i06/paper

Sen: https://onlinelibrary.wiley.com/doi/abs/10.1002/hyp.7224

Svensson et al. 2015. https://iopscience.iop.org/article/10.1088/1748-9326/10/6/064006

Harrigan et al. 2018. https://iwaponline.com/hr/article/49/2/552/37838/Designation-and-trend-analysis-of-the-updated-UK

Labat, 2010. https://www.sciencedirect.com/science/article/pii/S0022169410001071

Laize & Hannah 2010. https://www.sciencedirect.com/science/article/abs/pii/S0022169410003331?via%3Dihub

Mills: https://rmets.onlinelibrary.wiley.com/doi/epdf/10.1002/joc.1003

---

## Author Response (AR1)

We would like to thank Anonymous Referee #1 for their detailed review comments. We found them to be insightful, and, through our responses to them set out below, we believe that they have resulted in a much-improved paper.

**General Comments**

**General Comment #1**: In this paper the authors draw upon an extensive UK hydro-climatological data-set, comprising >700 gaged records of streamflow and estimated precipitation to examine 7-year periodicities related to the NAO. The range in catchment responses (and potential additional cycles at 2 and 5 years) are interesting and I think the paper is appropriate for this journal (with suggested revisions noted below), although I'm not personally convinced that this paper provides 'critical process understanding' as suggested in the abstract, and it's not clear how this work can translate to improvement of the practice and policy of water resources management (as suggested in the abstract).

**Response to General Comment #1:** We agree that the wording of 'critical process understanding' is too strong given the scale of the study. We would soften these statements to reflect the general trends we are asserting throughout the paper; this will also help to address the confidence we have in inferring hydrological pathways from the high-level catchment descriptors used (BFI and GRT). Furthermore, we agree that there needs to be further discussion around the translation to practice and policy and will add a paragraph to the discussion to account for this.

We have updated at lines 566-576 and 571- 579 to address this comment.

**General Comment #2:** For a non-UK reader it would be useful to provide more context in introducing the work and commenting on confounding influences (e.g. there is no mention / discussion of abstraction, nor the potential for anthropogenic water use / return flows, to influence streamflow). Moreover, the authors only mention one aquifer (the Chalk) and it is difficult to interrogate the results to consider the potential for varying periodicities on other geologies (and catchments with differing land use), given the scale of figures such as Fig. 5. At this scale of analysis, I would also question whether the authors are able to infer hydrological pathways with confidence (although the Groundwater Response Time concept appears very interesting).

**Response to General Comment #2:** We agree that the role of confounding influences has not been discussed sufficiently. We note that over the period of analysis there have been both changes in regulatory and water resource management practices and that the latter will not have been applied in a consistent manner over all the catchments. Given this we might expect anthropogenic effects to add noise to the observations, but that there is no reason to expect that they should impart a systematic signal or bias to the data and so systematically effect the observations or results. Counter to this, there is a president in literature for exacerbation of climatic cycles by anthropogenic processes which may affect the amplitude of cycles but again we would not expect any bias or confounding signal. We will add text to the data and discussion sections to highlight both of these points.

Furthermore, we understand that, since our results only show signals in areas dominated by Chalk, it is difficult to interpret pathway processes for other geologies. As such, and in order to avoid over-interpretation of the results, we will add cautionary statements to the results to highlight the focus on the Chalk and that these relationships may not be translatable to other geologies (with different land uses).

We have updated text at lines 146 – 151 and 444 – 450 to address this comment

**Specific Comments**

**Specific Comment #1:** There are multiple (>10) references missing in the ref. list including: Bloomfield & Marchant, 2013; Dixon et al. 2013; Forootan et al. 2018; Haarsma et al. 2015; Tanguy et al. 2016.

**Response to Specific Comment #1:** These will be fixed, and all citations will be properly listed in the reference list

The reference list and citations throughout the paper have been reviewed and updates (reference list at lines 634 – 937)

**Specific Comment #2:** It would be useful to include more specific details in the abstract.

Response to Specific Comment #2: We will update the abstract to give a better overview of the specific findings and outcomes

We have updated the abstract at lines 28 – 33 to add more specific detail form the results and discussion

**Specific Comment #3:** A stronger argument to demonstrate the effectiveness of the BFI in relating 'physical catchment processes to streamflow' would be helpful (e.g. as I understand the Bloomfield et al 2009 study, cited in support, focused on the Thames catchment).

**Response to Specific Comment #3:** We agree with this comment, and we will include a broader look at literature to demonstrate the effectiveness of BFI as an indicator for hydrological pathway dominance.

We have added a wider literature review addressing the use of BFI as an indicator at lines 172

**Specific Comment #4:** Lines 179-183 should be reworked (e.g. GRT is lowest in southern and eastern England, but highest in south-east England?).

**Response to Specific Comment #4:** Agreed, these lines will be reworded to improve clarity

We have amended lines 199-201 to improve the clarity of the GRT description

**Specific Comment #5:** Figures 2 & 5: are rather small, and it would be useful to reproduce these at a larger scale.

**Response to Specific Comment #5:** These figures will be provided at a sufficiently high resolution to enable the journal to reproduce at a larger size

No update to the text has been made following this comment.

**Specific Comment #6:** Is it possible to combine Figures 3 & 4 so the plots can be compared more easily?

**Response to Specific Comment #6:** We have tested this previously, but unfortunately it makes for a cumbersome plot which is more difficult to interpret. Following from this comment and others, we will update Figures 3 and 4 to more clearly show the spread of periodicity strengths.

No update to the text has been made following this comment.

**Specific Comment #7:** The Discussion is rather long and would benefit from a more selective review of the results, with more attention to suggestions for future research.

**Response to Specific Comment #7:** We agree that the discussion is long and following the comments from Anonymous Referee #1 and #2, we will seek to focus more on the discussion of hydrological pathways and less on the potential processes behind the results in the Winter and Summer rainfall data. This will also allow room for an improved discussion of policy and future work.

We have trimmed the superfluous discussion, and improved brevity to focus the discussion toward hydrological processes at lines 456-474, 511-515, 596-579; allowing for improved discussion of practice and policy.

**Anonymous Referee #2**

We would like to thank Anonymous Referee #2 for their detailed review comments. We found them to be insightful, and, through our responses to them set out below, we believe that they have resulted in a much-improved paper.

**General Comments**

**General Comment #1:** "I do struggle to fully grasp the significance of the work, however. The wavelet analysis reveals intriguing patterns but these periodicities are difficult to interpret. I find this paper like many similar wavelet/frequency papers in that a lot is down to the interpretation of the plots and assumptions made on driving processes - there is a lot of faith here seems to be on 'NAO like' signals which are taken and run with but to me are far from clear in the plots, and wrt previous work. There is no actual statistical link with the NAO (cf. FrancoVilloria, etc) which should be made clear from the outset (possibly even title?)."

**Response to General Comment #1:** We agree that there needs to be improved narrative around the reasons we are suggesting the detected 7-year cycle is the result of a widespread control such as the NAO (i.e. wide spatial domain that matches the known control of the NAO; common 7-year cycle found in both rainfall and streamflow). However, we also understand that we have not made any statistical comparison to NAO in this paper and therefore we are at risk of overinterpreting these results. As such we will add cautionary statements to the paper to highlight this and briefly discuss other possible causes for a widespread 7-year cycle. Furthermore, we would remove reference to the NAO in the title of the paper to avoid this bias and adjust the title to "Exploring the role of hydrological pathways in modulating multi-annual ocean-atmosphere teleconnection periodicities from UK rainfall to streamflow"

Claims of linkages between signals in the NAO and those found in streamflow and rainfall in this paper have been softened and caveated, including updating the title. Changes to the text to this affect are principally found at lines 37 – 77; 100; 125 – 128; 257 – 260; 428-441; 483-486

**General Comment #2:** As far as I understand the NAO literature it is somewhat debatable as to whether such patterns are physically meaningful. They do crop up in some papers but I have not seen so much that is convincing as to plausible physical interpretations (this can be said about a lot of work on cyclicities in hydroclimatic series in general). Most studies indicate that if NAO cyclicity exists it is weak, sensitive to choice of NAO index, season (DJF, DJFM) and tends to come and go over time as noted in the original Hurrell paper (these nonstationaririties being another barrier to practical applications; see also Franco-Villoria on nonstationarity of NAO-flow linkages). This should be elaborated on. See also important statistical work of Mills (2004). There is some discussion of some of this literature in the current paper (a lack of references in the reference makes it hard to find some of the key papers cited in relation to NAO periodicity) but I find some claims quite tenuous as to what the cited literature reveals wrt the findings of Rust et al. The discussion is very wide ranging, but at times highly speculative and overreaches beyond the analysis made here, and as noted below sometimes reflects misinterpretation of existing papers, or reference to work that is not directly relevant.

**Response to General Comment #2:** We agree that the nonstationary of the interpreted period relationship between the NAO and UK streamflow requires further discussion in the narrative of this paper, and that discussion is also required as to the question of whether the periodicities in the NAO are physically meaningful. We will supplement the current discussion to address this and to add levity to our assertions that the 7-year cycle may be the result of the NAO, and any resultant implications for practical application.

We have added text at lines 52 – 65; 71 – 73 and 76 to focus less on the periodicities of the NAO, and to highlight that many papers agree that the NAO is pseudo-periodic.

**General Comment #3:** Finally, I am not sure of the practical significance of such findings. The authors suggest this reveals a 'degree of forecasting' and 'critical process understanding' but I am not sure how water managers can really use such information beyond a general raised awareness. And for that, there are already operational physically based seasonal to decadal prediction systems (the Met Office DePreSys system being one) which already give an indication of potential NAO states (and moreover, more importantly, the large-scale variables that drive the NAO, like ENSO, QBO etc which are much more predictable). How would findings really be used in reality? Perhaps the authors could comment. That said, there are wider applications of these findings in explaining general time series evolution (trends and variability) a huge area of interest in the literature.

**Response to General Comment 3:** This has also been highlighted by Anonymous Referee #1. We agree that there is a need to soften these statements about 'critical process understanding', and indeed 'degree of forecasting' to take account of the high-level scale of the study and that we are documenting general tendencies, rather than specific physical processes. We will soften these claims throughout the paper. Furthermore, we will improve the narrative around the practical implications; that our identification of an NAO-like influence on streamflow (particularly summer low flows given BFI relationship) provides a practical purpose for water managers to use the NAO forecasting systems. Rather than focusing on NAO prediction, but demonstrating where an understanding of the NAO can be helpful in forecasting hydrological change.

We have softened the claims of providing critical process understanding at lines 32 and 52 – 77, and included more discussion as to the practical application of the findings at 571 - 579

**Specific Comments**

**Specific Comment #1:** L18 – 'degree of forecasting' – I see this word is being used in a general sense but I don't think this is really what the paper is offering up, rather some general tendencies of multiyear river flow behaviour in time series, itself possibly useful but not really 'forecasting', although perhaps for general preparedness. I think any findings from studies like this are more useful in providing context for studies of long-term trends and variability (helping shed light on widely reported 'flood poor', 'drought rich' periods and so on) rather than providing any information on preparedness. The authors could comment on this potential application too.

**Response to Specific Comment #1:** We agree with this comment that 'degree of forecasting' infers a quantitative prediction of future river flow variability. We will adjust the text to discuss the benefits of these cycles in improving understanding of flow behaviour in general and focus on preparedness for extremes, rather than referring to forecasting specifically.

We have updated the text at lines 19 – 20 and to soften these claims.

**Specific Comment #2:** L42 – L44 (around) maybe also worth clarifying early on that the NAO is primarily a driver of wintertime variation, noted later but should be in the intro.

**Response to Specific Comment #2:** Agreed – we will update the text in the introduction accordingly.

We have updated the text at lines 42-43 to address this comment.

**Specific Comment #3:** L48 – [and into discussion]. I wanted to look into past research on claims of NAO cyclicities but found it a bit lacking in these papers – Tabari doesn't really look at multiannial cycles; Su found cycles but did not appear to look at the NAO. Neves relevant and useful. Kuss and Gardak not in reference list; Meinke looked at ENSO; no reference in list for Tremblay; no reference in list for Olsen, but found it and appears to be a different beast on Paleo timescales so not sure of its relevance here. It's difficult to examine the wider evidence based with these missed references. Some that are cited appear to be not entirely directly comparable – I'd recommend a careful re-reading and re-positioning of this work with the literature here (and in the discussion]. See also missed references in intro and other useful international papers e.g. Labat (2010) on various possible mechanisms of cyclicity globally.

**Response to Specific Comment #3:** We agree that this comment that this work needs to be placed more carefully within the context of existing research; specifically, the presence of NAO cycles and their teleconnections. This follows on from Comment #2, and comments made by Anonymous Referee #1. We will adjust the introduction literature review and the discussion to make clear the fact that existing research is unclear as to the presence of cycles in the NAO and their impact on North Atlantic hydrology / meteorology. We will also ensure that the reference list is updated and complete.

We have updated the text and citations at lines 37 – 50, and added text at lines 445 – 451 in the discussion to address this comment.

**Specific Comment #4:** L67, L68 – there are many more studies that look at NAO influences on UK streamflow in winter and more generally and these shed important light on regional patterns in some detail, and also catchment characteristics – would be worth referring to these. See e.g. Laize et al. 2012, Svensson et al. 2015 and references in both. Laize &

Hannah really looked at this propagation question in some detail (for interannual rather than periodic behaviours but still relevant).

**Response to Specific Comment #4:** Thank you for highlighting these studies, we will review these and review our literature discussion accordingly.

We have updated lines 48 and 59 to included more of these references.

**Specific Comment #5:** L136 – L139. At 705 catchments, this will inevitably be a very mixed set of catchments subject to all sorts of human influences. Ideally hydroclimate studies linking large-scale drivers to hydrological response should use relatively undisturbed catchments - human disturbances can alter the chain of propagation from signal to streamflow response. I agree however that there are few such catchments (see e.g. Harrigan et al. 2018) which would limit the range of BFI/GRTs, so using the wider set is reasonable. While I expect that the outcomes from the broad-scale national picture is unaffected, there will likely be significant heterogeneity in the catchment results. This warrants a comment.

**Response to Specific Comment #5:** We agree that the influence of heterogeneity of catchment properties and the role of confounding influences in general needs a wider comment in the paper. We will highlight that over the period of analysis there have been both changes in regulatory and water resource management practices and that the latter will not have been applied in a consistent manner over all the catchments. Given this we might expect anthropogenic effects to add noise to the observations, but that there is no reason to expect that they should impart a systematic signal or bias to the data and so systematically effect the observations or results. We will add text to the discussion to this end, in addition to highlighting the potential influence of anthropogenic modulation of climate teleconnections.

We have updated the text at lines 147 – 153 to better justify our dataset and our expectation for any impact on the results.

**Specific Comment #6:** L163. This categorisation into even BFI classes will lead to very uneven numbers in the groups, given how skewed (and slightly bimodal) the BFI distribution is. Being that there is nothing meaningful about these arbitrary thresholds why not try and make them cover the distribution better? There are very few catchments at BFI <0.25, and not so many >0.75, with the great majority in the middle two classes. (I do not have this to hand but there are NRFA BFI distributions available somewhere in the literature I am sure and it would be easy to check). How would a more even classification effect the later results?

**Response to Specific Comment #6:** We agree that the distribution of sites in each category of BFI is not normal and therefore may lead to a skew in the results. This is however, mitigated by the Mann Whitney U significance testing in the final relationship testing which we draw our conclusions. We have previously tested equally sized bins and this did not change which relationships were signification, their direction or the final outcomes of the analysis. The current bins were chosen to allow us to test a spread of BFI values in their relationship with signal propagation (i.e. understanding the influence of increasing BFI from low baseflow proportions); which we would lose by generating equally-sized bins. However, we appreciate that the rationale for this was not discussed in the paper. We will highlight the assumptions implicit in the categorisation we have chosen, and improve the narrative around our reasons for choosing this categorisation.

We have added lines at 178 – 182 and 290 – 291 to address this comment and highlight the use of our significance testing.

**Specific Comment #7:** L184. The point here is a valid one that BFI is simply an empirical property from the baseflow, but this could be generalised – rather than referring to throughflow, it can reflect any terrestrial storage in soils, lakes (lake and reservoir catchments also have high BFI) and so on.

**Response to Specific Comment #7:** Agreed, we will adjust the text here to generalise the conceptual model of BFI as a representation of hydrological pathways.

We have improved our literature citations for the use of BFI as a representation of hydrological pathways here.

**Specific Comment #8:** L187 – I do not know how GRT is distributed but mirroring my BFI Comment above can the authors comment on this? It looks like it is very uneven from Fig 7 with many more in the highest class. Fig2 caption – might need to explain why this is log GRT as you have not introduced that yet, only referred to the categories.

**Response to Specific Comment #8:** We agree that more explanation is required for the specific groupings for GRT. Similar to BFI the groupings of GRT have been specifically chosen to investigate a range of GRT that are relevant to the propagation to these signals according to existing literature (such as Townley 1995). We will add text to the Methodology section to highlight this, and that our chosen categories are not equally populated.

We have added text at lines 294 - 295 to address this comment.

**Specific Comment #9:** L204 – just a comment really for future work. This study does not consider the transition seasons, which is fine. But if the focus is really on propagation from winter NAO met signals I would have thought looking at spring would be really interesting – it may help better explain the propagation from winter.

**Response to Specific Comment #9:** Noted and we thank the Referee for their insight regarding this. For the purposes of this paper, we want to focus on understanding a general trend between baseflow contribution and NAO-like signal presence. As such we chose to test between winter and summer flow to obtain the greatest difference between baseflow contributions (given the perennial nature and temperate climate of all the catchments within the study).

No update to the text has been made following this comment.

**Specific Comment #10:** L250. (and 2.3.4 generally). I'm not sure I fully follow the logic here so it needs some clarification – at the moment it sounds like this process is doing some heavy lifting. I failed to follow the process leading up to L250 that indicates "this produced a wavelet power for each dataset that is considered NAO like". Why would this be considered NAO like a priori? This sounds particularly important given the 'NAO like' signal is then used to producing the residual series that is so important thereafter for capturing the 'measure of modulation of signal strength'.

**Response to Specific Comment #10:** We agree that further detail is needed to explain the rationale for selecting the representative 7-year periodicity from the dataset and indeed the a priori 'NAO-like' description. The intention is to take an a priori understanding that any wide-spread multi-annual periodicity found in both rainfall and streamflow around the 7-year range is likely the result of the NAO, given the NAO's wide spread control and assertions of previous research that the NAO exhibits a weak ~7-9 year periodicity (although we agree that we need to soften these claims and add cautionary text to explain that existing research is undecided as to whether the cycles in the NAO are meaningful). We agree that this narrative is not clear and that further text is needed to clarify that no statistical relationship

between these cycles and the NAO is drawn. This text will be added to the Methodology and to the Discussion.

We have added text at lines 265-278 and amended text at lines 428 – 453 to better reflect our use of the wavelet spectra as indicators of the common, wide-spread multi-annual periodicities in UK streamflow.

**Specific Comment #11:** L258. Following the above, it also appears to be a big assumption to produce this residual series for the summer based on winter rainfall. While in very high BFI catchments a very long lag time may be expected a priori, this is not necessarily the case in many (most?) cases (See my earlier comment about the spring season being omitted). I'm just concerned that two comments (while partly no doubt reflecting my lack of understanding of the process) are reflective of some major assumptions being made in this study which are (at face value) in danger of baking in some of the conclusions, somewhat – hopefully a clearer exposition will help allay these fears.

**Response to Specific Comment #11: O**ur rationale for following this method is that (due to the perennial nature of all UK catchments in this study) even in catchments with lower BFI, variability in winter rainfall (in the dominant period of groundwater recharge) will influence summer flows that are generally baseflow-dominated. Where catchment have very little baseflow contribution, summer flows and their periodic behaviour are minimal and so our residuals simply show a lack of periodic behaviour in these instances. We will add text to the methodology section to clarify this aspect.

We have added text at lines 280 – 284 to better explain and justify this step.

**Specific Comment #12:** L271 - Given this pairwise testing between groups, is my earlier comment about the irregular distribution of BFI a potential issue (would this look different with different more representative groups?)

**Response to Specific Comment #12:** We agree that the distribution of sites in each category of BFI is not normal and therefore may lead to a skew in the results. Our pairwise significance testing for these groups is the Mann Whiney U test which is non-parametric and is appropriate for non-normally distributed data. As such this issue is mitigated by the significance testing. Additionally, these bins were chosen to allow us to test a spread of BFI values in their relationship with signal propagation.

We have added text at lines 178 – 182 and 290 – 291 to address this concern.

**Specific Comment #13:** L285 I am no expert on wavelets. But when I look at Figs 3 and 4 I wouldn't say a c.7 year cycle leaps out at me – rather, higher powers at a range of years <c.8 years. Especially for winter rainfall. That is, I see there are peaks in significance but are you really that confident in there being a (even approx.) 7 year cycle in these? IN fact I don't really see the 'two discrete bands of periodicity' (l279). I guess this is all down to interpretation but this causes some modest concern if this is the basis of the identification of 'NAO like' signals – please elaborate on this. It's also really difficult to see the variability in the wider cloud of catchments shaded grey, but it looks like there is a range, especially for rainfall – some comment of this would be useful.

**Response to Specific Comment #13:** The purpose of Figure 3 and 4 is to show that the 7-year periodicity is apparent, especially in monthly and winter streamflow, and we agree that at present these do not do that sufficiently. We would suggest the individual catchment lines in these figures are made clearer to better show the range of periodicity strengths. Furthermore, we will alter the text in the results and the discussion to better highlight the

purpose of these figures (i.e. that we expect to see a band of increased strength and increased significance around the 7-8 year band in monthly and winter rainfall, if there was a periodic teleconnection between the NAO and rainfall, that propagates to streamflow), and that this aligns with periods found in previous research.

We have added text at lines 265 and 300 – 303 to highlight the intended purpose of these plots for the identification of common periodicities in UK streamflow and have added text to the discussion at lines 427 – 431 to reiterate these points. Furthermore we have amended Figures 3 and 4 to better show this multi-annual periodicitiy.

**Specific Comment #14:** L290. 'Wavelet p values indicate the detected wavelet powers are not the result of external forcing'. Is this strictly true, I thought this just indicates it is not AR1 generated – I assume it does not rule out that it is internal variability, which could be driven by all sorts of long-term persistence processes (see the extensive work on the Hurst phenomenon and many papers of Demetrios Koutsoyiannis), as opposed to external forcing. (corollary to this, re: my point in the general intro about physical significance, when I backtrack into the literature on NAO periodicities, back to the Hurrell papers cited, it seems far from clear as to being settled whether NAO periodicities, as they are, are externally forced or internal variability).

**Response to Specific Comment #14:** We agree that there is a need for clarity of internal vs external variability in the NAO. In this instance, we need to make clear that even if the NAO is only internally variably, this can produce a behaviour in rainfall and streamflow that is externally forced (and therefore what we are testing the significance of). We also agree that additional text is required to highlight that this AR1 test does not specifically mean it is the result of the NAO, but simply not entirely the result of internal noise of the rainfall / streamflow data. Text to clarify this will be added to the Methodology section and again highlighted in the Discussion section.

We have added text to lines 256-259 to highlight the intended use of the red noise testing, with regard to the NAO and the dataset in use.

**Specific Comment #15:** L294 – should this say 'river flow records' instead of groundwater?

**Response to Specific Comment #15:** Yes, this should be 'river flow records'. This will be corrected.

The text has been corrected at line 316

**Specific Comment #16:** L340 onwards and Fig 6 – note whether previous question of BFI distribution has any impact on these findings?

**Response to Specific Comment #16:** We agree that we need to highlight that the data it not normally distributed, however our choice of significance testing is suitable for non-normally distributed data therefore this will have minimal impact on the overall conclusions from this figure. We will, however, add text to highlight the non-normal distribution of the bins.

We have added text to lines 178 – 182 and lines 290 – 291 to address this non-normal distribution.

**Specific Comment #17:** L416 – I would not have said this strong conclusion on the difference between the periodicities between winter ('present') and summer ('absent') rainfall really emerges from Figs 2 and 3 as noted earlier. I may be missing something but this

seems quite an open interpretation of those data. Important as the seasonal differences are majored on.

**Response to Specific Comment #17:** We agree that the wording here could be improved and would propose changing to "Additionally, the stronger signal presence in winter compared to summer rainfall apparent in Fig 4 generally agrees with existing research showing that NAO's control over European rainfall is primarily expressed in winter months (Trigo et al., 2004; West et al., 2019)."

Added text at lines 439 – 440 to address this comment.

**Specific Comment #18:** L420 – L425. Following on from this, this discussion hinges on there being multiple periodicities at different peaks between the seasons, but my reading of those plots makes it hard to really pick out any of these as 'peaks'.

**Response to Specific Comment #18:** We agree that this paragraph doesn't progress the narrative of the overall paper. We would remove this paragraph and add to the previous paragraph stating that we will be focusing on the 7-year periodicity.

We have removed discussion of the noisier peaks in the spectra to focus our discussion on the most common and strongest multi-annual cycle in UK streamflow.

**Specific Comment #19:** L425 – I also think this 16 – 32 periodicity is very difficult to see, let alone link to the EA pattern – and I could find no work on this in the Rust et al. 2019 paper cited? An important general point though is that while the NAO is the leading mode of variability there is a whole laundry list of (interacting) influences (Scandinavia pattern, AO, etc) as well as the lower frequency SST drivers (AMO, ENSO) that are not considered here given focus on NAO-like, even though other work suggests they could also manifest themselves on similar timescales (see e.g. Labat, Villoria). Worth comment in intro & discussion?

**Response to Specific Comment #19:** We agree that the discussion around 16-32 periodicity is difficult to see in the figures provided, and that ultimately the discussion of the 16-32 periodicity is unnecessary to the overall narrative, meaning this paragraph feels unnecessary. However, we also agree that there needs to be a more robust discussion as to the confounding influence of multiple other climatic oscillations that have been shown to influence the NAO's control on UK weather, which further adds complications to the teleconnection. We would remove this paragraph and add text to the introduction that prefaces this work by highlighting the known confounding influences in climatic teleconnections and that to soften the links directly to the NAO again highlight the uncertainty in understanding these teleconnections.

We have added test at lines 443 – 449 to address these confounding issues

**Specific Comment #20:** L453. Haarsma not in the reference list, But on looking at this paper I don't see this SE England outcome on any of their results maps? Please clarify. In general, I find this whole gulf stream section really speculative. I looked (admittedly quickly) but could not really find much in these papers to support this e.g. concurrent but lagged correlations in Wavelet rather than periodicities; little specific mention of GS in Hurrel and Deser. A lot is made of the GS as a mechanism for the key NAO-like behaviour central to this study, so this reference to other work is important and could be checked and strengthened.

**Response to Specific Comments #20:** We would propose strengthening the literature review of this section and highlight that understanding the atmospheric – oceanic pathways

is not the purpose of this paper, and additionally highlighting the need for further work to be undertaken.

We have updated lines 456 – 474 to better cite existing literature and shorted this section to focus the discussion on hydrological processes

**Specific Comment #21:** L576 – L582. Related to my points in the intro, good to see potential applications but this is quite a long way off from what is discovered in this paper so some of these claims could be moderated.

**Response to Specific Comment #21:** Agreed, we will seek to moderate these claims and the slimier claims made elsewhere in the paper to take account of the high-level nature of the study, and highlight the need for further work to investigate these processes in more detail before this is used directly for screening purposes

We have adjusted the text at lines 586 – 588 and lines 593-594 such that the discussion of application better reflects the high-level nature of the study.